# Classification Method of Significant Rice Pests Based on Deep Learning

Zhiyong Li [1,2,†], Xueqin Jiang [1,2,†], Xinyu Jia [1,2,†], Xuliang Duan [1,2], Yuchao Wang [3] and Jiong Mu [1,2,*]

1   College of Information Engineering, Sichuan Agricultural University, Ya'an 625000, China
2   Sichuan Key Laboratory of Agricultural Information Engineering, Ya'an 625000, China
3   College of Mechanical and Electrical Engineering, Sichuan Agricultural University, Ya'an 625000, China
*   Correspondence: jmu@sicau.edu.cn; Tel.: +86-133-4060-8699
†   These authors contributed equally to this work and should be regarded as co-first authors.

**Abstract:** Rice pests are one of the main factors affecting rice yield. The accurate identification of pests facilitates timely preventive measures to avoid economic losses. Some existing open source datasets related to rice pest identification mostly include only a small number of samples, or suffer from inter-class and intra-class variance and data imbalance challenges, which limit the application of deep learning techniques in the field of rice pest identification. In this paper, based on the IP102 dataset, we first reorganized a large-scale dataset for rice pest identification by Web crawler technique and manual screening. This dataset was given the name IP_RicePests. Specifically, the dataset includes 8248 images belonging to 14 categories. The IP_RicePests dataset was then expanded to include 14,000 images via ARGAN data augmentation technique to address the difficulties in obtaining large samples of rice pests. Finally, the parameters trained on the public image ImageNet dataset using VGGNet, ResNet and MobileNet networks were used as the initial values of the target data training network to achieve image classification in the field of rice pests. The experimental results show that all three classification networks combined with transfer learning have good recognition accuracy, among which the highest classification accuracy can be obtained on the IP_RicePests dataset via fine-tuning the parameters of the VGG16 network. In addition, following ARGAN data augmentation the dataset demonstrates high accuracy improvements in all three models, and fine-tuning the VGG16 network parameters obtains the highest accuracy in the augmented IP_RicePests dataset. It is demonstrated that CNN combined with transfer learning can employ the ARGAN data augmentation technique to overcome difficulties in obtaining large sample sizes and improve the efficiency of rice pest identification. This study provides foundational data and technical support for rice pest identification.

**Keywords:** rice; pests; classification; datasets; transfer learning; ARGAN

## 1. Introduction

Rice represents one of the major global food crops for human consumption and is the cornerstone of world food security. The growth cycle of rice is accompanied by the occurrence of several pests that lead to serious yield losses. Accurate identification of rice pests facilitates timely preventive measures to avoid economic losses [1]. Early rice pest monitoring methods mainly use trapping lights to capture pests that are then retrieved, counted, and identified the following day. This labor-intensive process is performed manually for purposes of assessing the current pest situation, and also provides room for misdiagnosis and delays in diagnosis time. These factors directly affect the accuracy and timeliness of pest control [2]. With the continuous development of machine learning and deep learning, increasing numbers of scholars are attempting to extend these novel approaches to the field of pest identification.

Most of the previous work on pest identification can be attributed to the traditional machine learning classification framework, which consists of two main modules: (1) Feature

extraction module for pest images. That is, the whole image is represented by handcrafted features, including Gabor [3], HOG [4], GIST [5], SIFT [6], and SURF [7]; (2) Machine learning classifiers. This includes techniques such as support vector machines [8–10], Naive Bayes [11] and k-nearest neighbor (KNN) [12]. Such methods rely on the accurate extraction of feature parameters, and once any wrong features are extracted, it is difficult for machine learning classifiers to accurately identify pests with similar features.

Recently, deep learning techniques have attracted much attention from rice pest researchers [13–15]. Liu et al. [16] classified rice pests by training a deep CNN with their dataset covering 12 species and including about 5000 training samples. Alfarisy et al. [17] collected 4511 training samples using a search engine and used CaffeNet [18] for rice pest identification. Burhan et al. [19] comparatively studied the performances of five deep learning models (Vgg16, Vgg19, ResNet50, ResNet50V2 and ResNet101V2). Overall, these depth-feature-based efforts lack sufficient samples to optimize the large number of hyperparameters of CNNs. To address the problem of limited pest species and samples, Xiaoping Wu et al. [20] collected a large-scale IP102 dataset of eight pest-infested crops covering 102 species including a total of 75,222 samples. They evaluated the performance of state-of-the-art deep convolutional neural networks including AlexNet, GoogleNet, VGG16, and ResNet50 on the IP102 dataset, where ResNet achieved the best results in all indicators. However, the highest classification accuracy of only 49.4% indicated the challenging nature of the IP102 dataset.

Deep learning [21] is a method based on big data [22], and we certainly hope that the larger and higher quality the data are, the better the model generalization ability. At the same time, we hope the data can cover various scenarios. However, it is often difficult to cover all the scenarios when data are collected. This is where data augmentation [23] presents an effective way to expand the data sample size. However, the relatively small diversity and variability of the generated images using classical data augmentation has facilitated research on GAN data augmentation [24], and the samples generated by GAN introduce more variability and can further enrich the dataset to improve the accuracy of the training process [25]. Nazki et al. [26] proposed a framework for the data augmentation of plant disease datasets using GAN networks to address the problems of imbalance and lack of samples in the dataset, and demonstrated the effectiveness of synthetic data augmentation of GAN over classical data augmentation for image classification and detection tasks. Ding B et al. [27] proposed a focused attentive recurrent generative adversarial network (ARGAN), and experimental results proved that the method outperformed state-of-the-art methods. Haseeb Nazki et al. [28] introduced ARGAN, which differs from previous approaches, by optimizing Activation Reconstruction loss (ARL) and adversarial loss to render more visually compelling synthetic images of the dataset.

Considering the highly unbalanced nature of the IP102 [20] dataset and the challenge presented by the highest accuracy rate of only 49.4%, we reorganized a dataset for rice pest identification based on the IP102 dataset by Web crawler technique and manual screening, named IP_RicePests. Specifically, the dataset includes over 8248 images in 14 categories with a natural long-tailed distribution of data. In addition, we expand the IP_RicePests dataset to 14,000 images by ARGAN data augmentation technique and use the parameters trained on ResNet [29], VGGNet [30] and MobileNet [31] networks based on the public image dataset, ImageNet, as the initial values for network training to achieve image classification in the field of rice pests.

## 2. Related Work

Datasets form the basis for building deep learning models, and large-scale, high-quality datasets tend to improve the quality of model training and the accuracy of prediction. However, some existing datasets related to pest identification mostly contain no more than 1000 samples, such as those in [32–36] containing only 1440 samples covering 24 categories of common field crop pests, and with each category containing only 60 samples, a number which renders CNN model training a difficult task. To address this problem,

several larger datasets have subsequently emerged. Refs. [16,17,37,38] proposed several datasets containing more than 4500 samples, with 100 samples per category. Ref. [20] proposed an open source dataset IP102 containing 75,222 samples covering 102 classes of common pests of field crops and evaluated the classification performance using hand-designed features (including CH, LCH, Gabor, GIST, SIFT and SURF) and deep learning networks (including AlexNet, GoogleNet, VGGNet-16 and ResNet-50), respectively, all of which were pre-trained on ImageNet and then fine-tuned on the IP102 dataset. ResNet achieves the best results in all metrics, while the huge difference between 49.4% accuracy and 31.5% G-mean shows the highly unbalanced nature of the IP102 dataset. Moreover, the highest accuracy of only 49.4% indicates the challenging nature of the IP102 dataset. Therefore, we decided to continue to advance our research on the imbalance learning problem based on the IP02 classification system.

## 3. Materials and Methods

### 3.1. Image Acquisition

To facilitate further scientific research and practical applications, we wanted to address the problem of limited rice pest species and samples. Therefore, we compiled a large-scale dataset IP_RicePests for rice pest identification based on the classification system of IP102. We collected and labeled the dataset through the following three stages: (1) image collection, (2) image primary screening, and (3) professional data labeling.

In the image collection phase, we used the IP102 dataset as the main source for collecting rice pest images and combined it with Python web crawler technology to automatically collect a large number of images of 14 rice pests from several specialized agricultural and insect science websites. In the initial image screening stage, we organized 2 volunteers to manually screen the rice pest images obtained from the IP102 dataset as well as the web crawler. The volunteers removed images with no pests or with more than one pest in them. For example, Figure 1 shows some of the poor sample images in the IP102 dataset, which can cause different degrees of damage to the classification accuracy, and this may be one of the main reasons why the highest accuracy is only 49.4%. In the professional data annotation stage, we invited 1 expert with specialized knowledge on rice to annotate each image after the initial screening.

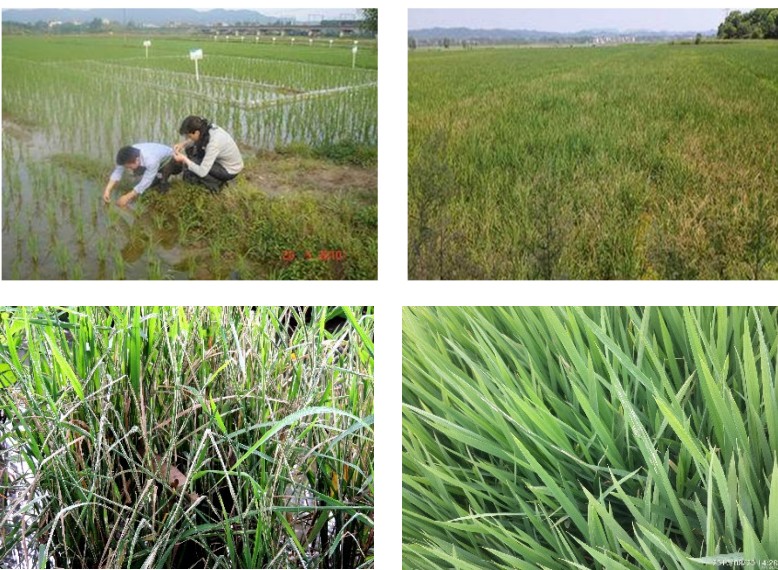

**Figure 1.** Images showing some of the poorer samples in the IP102 dataset.

Overall, the number of rice pest samples in IP_RicePests and IP102 remained largely consistent (as shown in Table 1).

**Table 1.** Comparison between the number of IP102 and IP_RicePests.

| Categories | IP102 | IP_RicePests |
|---|---|---|
| Asiatic rice borer | 1073 | 1000 |
| Brown plant hopper | 834 | 800 |
| Grain spreader thrips | 173 | 174 |
| Paddy stem maggot | 241 | 250 |
| Rice gall midge | 506 | 520 |
| Rice leaf caterpillar | 487 | 500 |
| Rice leaf hopper | 404 | 411 |
| Rice leaf roller | 1115 | 1110 |
| Rice shell pest | 409 | 401 |
| Rice stemfly | 369 | 370 |
| Rice water weevil | 856 | 860 |
| Small brown plant hopper | 553 | 556 |
| White backed plant hopper | 893 | 792 |
| Yellow rice borer | 504 | 504 |
| **Total Quantity** | 8417 | 8248 |

The IP_RicePests dataset included 8248 images covering 14 rice pest species (some images are shown in Figure 2). To obtain more reliable test results on IP_RicePests, there should be enough samples for each category in the test set. Therefore, we divided the dataset into training, validation, and test sets at an approximate ratio of 6:2:2. Specifically, IP_RicePests used for the classification task was divided into 4950 images for training, 1649 images for validation and 1649 images for testing.

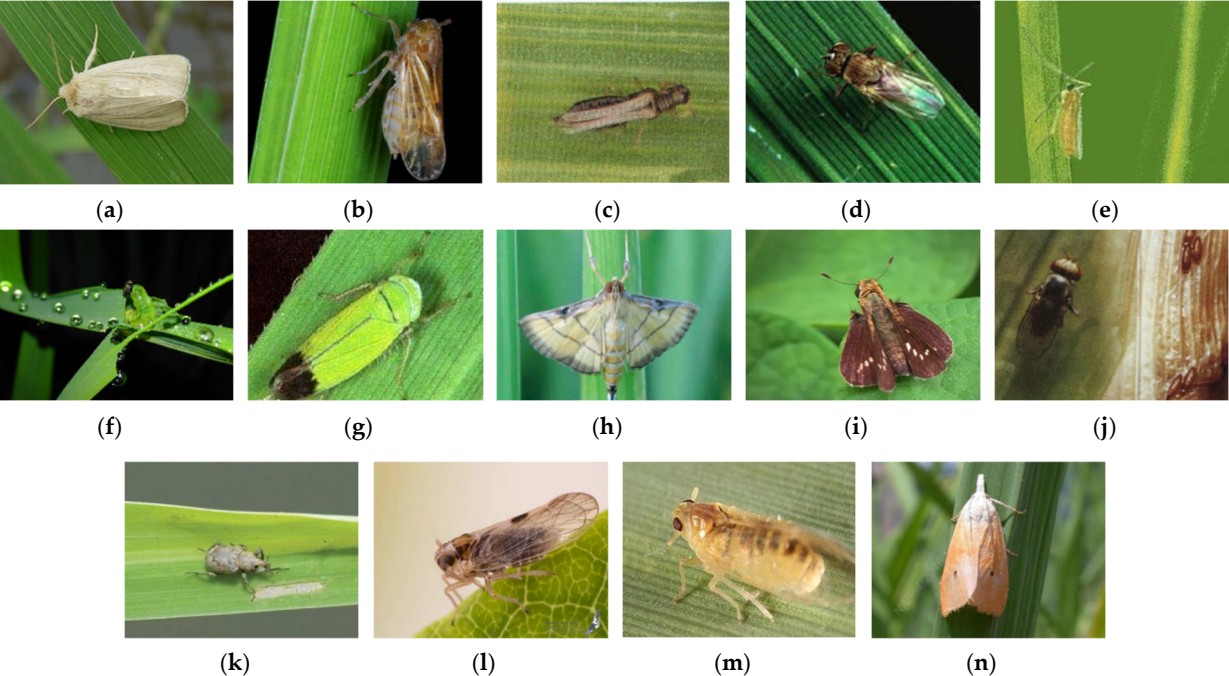

**Figure 2.** Example images from the IP_RicePests datasets, each image corresponds to a different species of rice pest: (**a**) Asiatic rice border; (**b**) Brown plant hopper; (**c**) Grain spreader thrips; (**d**) Paddy stem maggot; (**e**) Rice gall midge; (**f**) Rice leaf caterpillar; (**g**) Rice leaf hopper; (**h**) Rice leaf roller; (**i**) Rice shell pest; (**j**) Rice stemfly; (**k**) Rice water weevil; (**l**) Small brown plant hopper; (**m**) White backed plant hopper; (**n**) Yellow rice borer.

As can be seen from Figure 2, the IP_RicePests dataset includes 14 types of pests, including Asiatic rice border, brown plant hopper, grain spreader thrips, paddy stem maggot, rice gall midge, rice leaf caterpillar, rice leaf hopper, rice leaf roller, rice shell pest, rice stem-

fly, rice water weevil, small brown plant hopper, white backed plant hopper and yellow rice borer. The largest category contained 1110 samples (rice leaf roller) and the smallest category had only 174 samples (grain spreader thrips). Figure 3 shows the distribution of the number of samples for each pest category in the IP_RicePests dataset, which shows that the dataset exhibits a natural long-tailed distribution, and clearly demonstrating that IP_RicePests has a high imbalance rate (IR) [39] in most categories. This is mainly due to the complexity of the rice field environment and other reasons that make it difficult to collect samples of individual rice pests, such as grain spreader thrips, paddy stem maggot, rice stemfly and other pests, and the number of samples therefore tends to be low. Unbalanced data can lead to biases in the classification model learning in classes with relatively more training samples, and therefore, unbalanced data distribution should not be ignored.

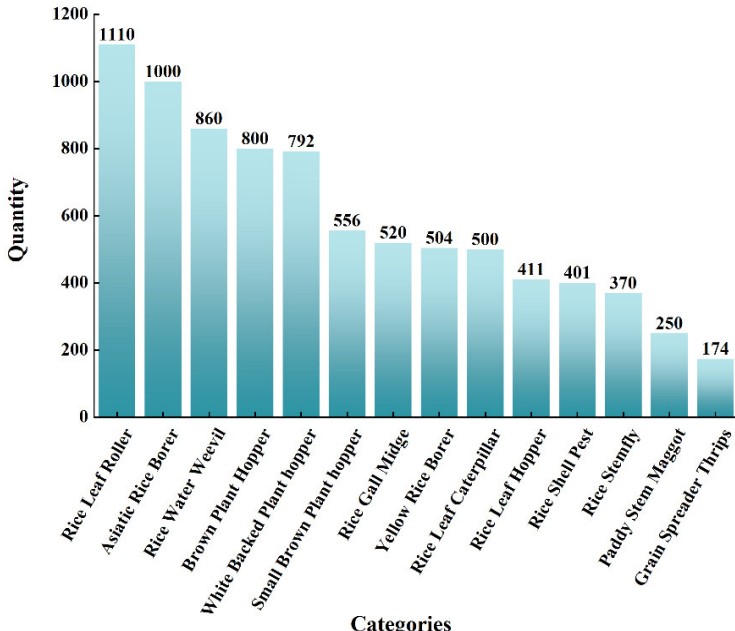

**Figure 3.** Distribution of the number of samples in each category of the IP_RicePests dataset.

### 3.2. Transfer Learning

Transfer learning [40] refers to the application of knowledge or patterns learned on a domain or task to a different but related domain or problem. The core idea is to transfer labeled data or knowledge structures from related domains, and accomplish or improve the learning effectiveness of the target domain or task. For convolutional neural networks, transfer learning enables pre-trained convolutional neural networks to be retrained with a small number of datasets, yielding better results compared to training from scratch. In some cases, for deep neural networks, there may not be enough data to train the network. Two common transfer learning strategies in deep learning are deep feature extraction and fine-tuning, respectively. In this experiment, we decided to take a parameter-based transfer learning approach to fine-tune the last three layers of the network: the fully connected layer, the SoftMax layer, and the classification layer. After replacing the new layers, the CNN model is then trained according to the new dataset IP_RicePests [34].

In the experiments of this paper, we chose the most common and widely used ImageNet dataset [41] to pre-train the model, which is currently the largest database globally for image recognition and is of great significance in the fields of image classification and target detection. In addition, the learning results are transferred to the model using the IP_RicePests dataset, which can improve the generalization ability of the model to some extent. The model demonstrates good transfer learning classification ability even under complex natural conditions.

### 3.3. ARGAN Data Augmentation

In deep learning, the number of samples is generally required to be sufficient, the greater the number of samples, the better the trained model effect and the stronger the generalization ability of the model. However, in practice, the number of samples is insufficient or the quality of samples is not good enough, which requires data augmentation of samples to improve the quality of samples. The purpose of data augmentation is to increase the variability of the input images so that the designed model bears greater robustness to images obtained in different environments. For a dataset with unbalanced samples, for example, if the number of images in a category is too small, data augmentation can increase the number of this category. Therefore, for the problem regarding IP_RicePests having an unbalanced number of classes, this paper uses the AR-GAN network for data augmentation.

AR-GAN is suitable for training and employs a semi-supervised strategy to make full use of sufficient unsupervised data, and its core idea is to introduce an activation reconstruction module based on CycleGan and to calculate the activation reconstruction loss ARL of $a$ and $G_{AB}(a)$ [42,43]. This enhances the perceptual realism of the real and generated images and allows the dataset to present more visually compelling synthetic images. The framework of AR-GAN is shown in Figure 4, where $A$ and $B$ are two types of rice pests. The generator $G_{AB}$ (or $G_{BA}$) converts the images from one category $A$ (or $B$) to another category $B$ (or $A$). Each category has a corresponding discriminator, such as $D_A$ for category $A$ and $D_B$ for category $B$. The purpose is to determine whether the image belongs to that category. Additionally, the L1 loss between $a$ (or $b$) and the reconstructed input $G_{BA}(G_{AB}(a))$ (or $G_{AB}(G_{BA}(b))$) achieves self-consistency.

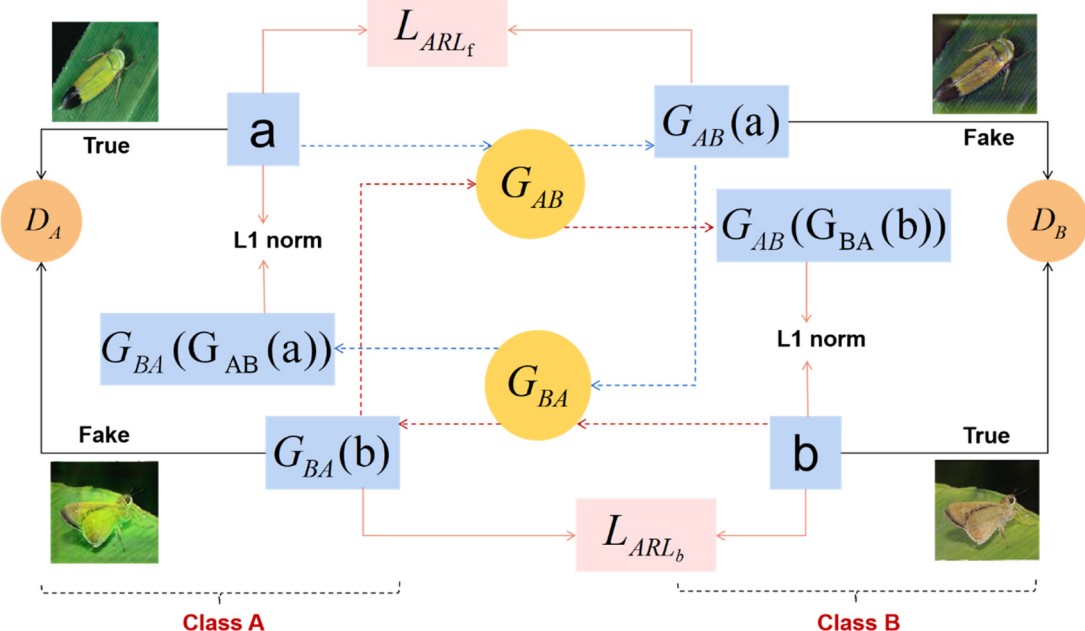

**Figure 4.** The framework of AR-GAN.

The principle of AR-GAN is shown in Equation (1):

$$\zeta_{total} = \zeta_{cycleGAN}(G_{AB}, G_{BA}, D_A, D_B) + \lambda\zeta_{ARL} \tag{1}$$

where $\lambda$ is the hyperparameter of the conditioning term, the gradual increase of which in this experiment improves both the aesthetic and perceptive quality of the model, and also refers to the cycle consistency loss [44], which can be expressed as:

$$\zeta_{cycleGAN}(G_{AB}, G_{BA}, D_A, D_B) = \zeta_{GAN}(G_{AB}, D_B) + \zeta_{GAN}(G_{BA}, D_A) + \alpha\zeta_{cyc}(G_{AB}, G_{BA}) \tag{2}$$

$$\zeta_{GAN}(G_{AB}, D_B) = E_{bP_{B(b)}}[logD_B(b)] + E_{aP_{A(a)}}[log(1 - D_B(G_{AB}(a)))] \tag{3}$$

$$\zeta_{cyc}\left|\left|(G_{AB}, G_{BA}) = E_{aP_{A(a)}}G_{BA}(G_{AB}(a)) - a\right|\right| + \left|\left|E_{bP_{B(b)}}G_{AB}(G_{BA}(b)) - b\right|\right| \tag{4}$$

In addition, the activation reconstruction loss [42,43] aims to enhance the perceptual realism between the real and generated images and improve the stability of the model [28]. It is defined as:

$$\zeta_{ARL} = \frac{1}{m}||A_a^n - A_b^n||_F^2 \tag{5}$$

where $||\hat{A}||_F$ denotes the Frobenius norm, $m$ is the shape of the feature map, and $n$ is the nth layer used in the feature extraction network.

### 3.4. Evaluation Indicators

The IP_RicePests dataset contains an unbalanced distribution of categories. We used several composite metrics commonly used for classification tasks, including Accuracy, Precision, Recall, and F1-score. Accuracy is the most common evaluation indicator and measures the ratio of the number of all correctly predicted samples to the total number of samples. Precision describes the ability of the classifier to not mark negative samples as positive. Recall indicates the ability to find all positive samples in a given category. F1-score balances the accuracy and recall. The expressions for Accuracy, Precision, Recall and F1-score are as follows:

$$Accuracy = \frac{TP + TN}{TP + TN + FP + FN} \tag{6}$$

$$Precision = \frac{TP}{TP + FP} \tag{7}$$

$$Recall = \frac{TP}{TP + FN} \tag{8}$$

$$F1 - score = \frac{2 \times Precision \times Recall}{Precision + Recall} \tag{9}$$

where $TP$ is the number of correctly predicted positive samples, $TN$ is the number of correctly predicted negative samples, $FP$ is the number of incorrectly predicted positive samples, and $FN$ is the number of incorrectly predicted negative samples.

### 3.5. Training Environment

The experiments were run on a computer with Windows 10 operating system, an Intel Core i7-9700 CPU with 16 GB memory, an NVIDIA GeForce RTX2060 graphics card, and software environment CUDA10.1, python3.7. The AR-GAN data augmentation task and training of each classification model were based on the PyTorch 1.7.0 framework. After several experiments, the hyperparameter settings in Table 2 were obtained.

**Table 2.** Model Parameters.

| Parameters | ARGAN | ResNet50/VGG16/MobileNet (Freeze Process) | ResNet50/VGG16/MobileNet (Unfreeze Process) |
|---|---|---|---|
| Batch Size | 8 | 4 | 8 |
| Epoch | 100000 | 100 | 100 |
| Optimizer | Adam | Adam | Adam |
| Learning rate | 0.0001 | 0.001 | 0.0001 |

## 4. Results

Deep learning features are effective for image classification. In this section, we evaluated the performance of deep convolutional neural networks on the IP_RicePests dataset, including ResNet-50, VGG16 and MobileNet. All networks were pre-trained on ImageNet

and then fine-tuned on the IP_RicePests dataset. To fully evaluate the IP_RicePests dataset, we first evaluated the classification performance of several deep convolutional neural networks on the IP_RicePests dataset. Subsequently, we evaluated the classification performance of several deep convolutional neural networks on the IP_RicePests dataset with ARGAN data augmentation.

### 4.1. Parameter Fine-Tuning of ResNet, VGGNet and MobileNet on ImageNet Dataset

In this experiment, transfer learning was introduced and the ImageNet dataset [39] was selected to pre-train the model and the learning results were transferred to the model trained using the IP_RicePests dataset. The loss curves of the training and validation sets before and after the introduction of transfer learning for the three classification models of ResNet, VGGNet and MobileNet are shown in Figures 5–7. From the curves, it can be seen that using the fine-tuned transfer learning strategy can obtain lower initial training loss and validation loss than training from scratch, and the loss function value decreases faster during the training process after the introduction of pre-training weights. Additionally, the whole curve shows a smoother convergence trend. In particular, Figure 6 clearly shows that the introduction of transfer learning effectively solves the overfitting problem of training the VGG16 classification network model from scratch.

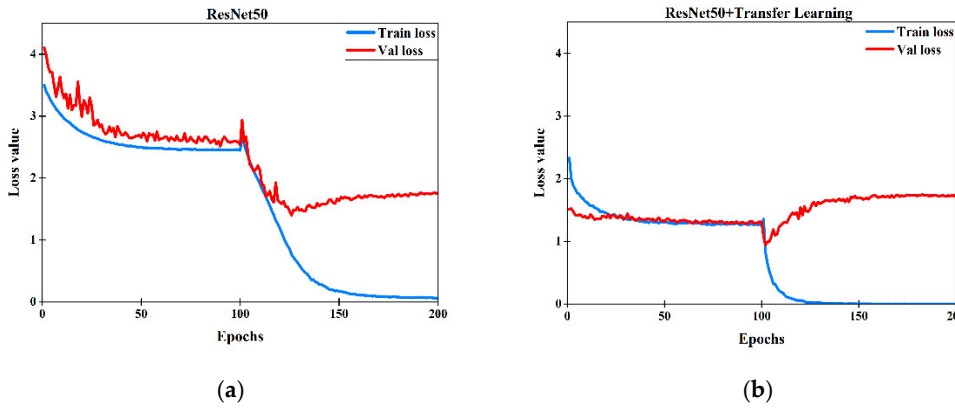

(a)                                                        (b)

**Figure 5.** Training loss and validation loss curves before and after the introduction of transfer learning in ResNet50. (**a**) ResNet50 trained from scratch; (**b**) ResNet50 with Transfer learning introduced.

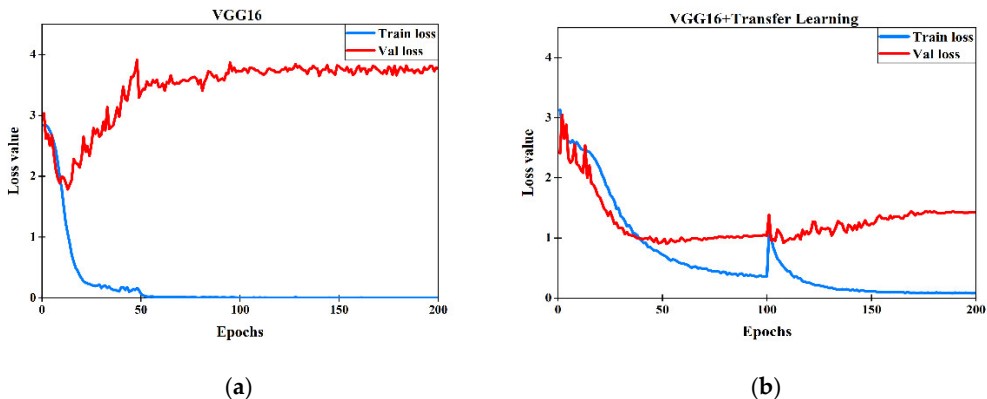

(a)                                                        (b)

**Figure 6.** Training loss and validation loss curves before and after the introduction of transfer learning in VGG16. (**a**) VGG16 trained from scratch; (**b**) VGG16 with Transfer learning introduced.

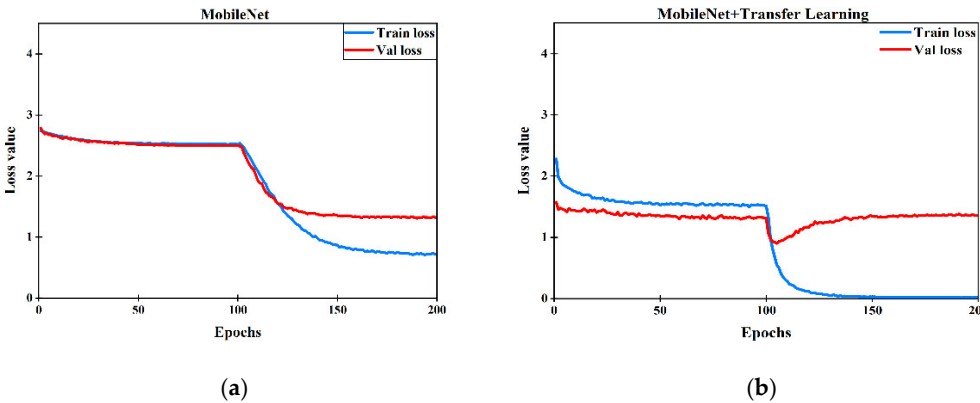

**Figure 7.** Training loss and validation loss curves before and after the introduction of transfer learning in MobileNet. (**a**) MobileNet trained from scratch; (**b**) MobileNet with Transfer learning introduced.

Figure 8 displays the accuracy curves of the validation sets of ResNet, VGGNet and MobileNet networks before and after using the transfer learning strategy in turn. It shows that the initial validation accuracy of the ResNet50, VGG16, and MobileNet classification network models increases after the introduction of transfer learning, and the training process curves using transfer learning converge better and more smoothly. It is worth mentioning that the validation accuracy of the network models after the introduction of transfer learning is always higher than that of the network models trained from scratch during the whole process.

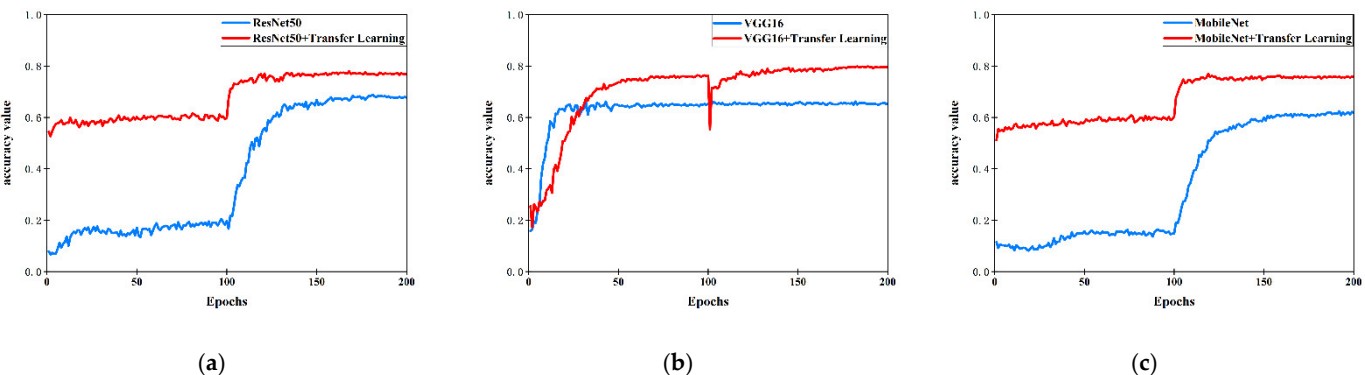

**Figure 8.** Accuracy curves of the validation set before and after using transfer learning for ResNet50, VGG16 and MobileNet. (**a**) ResNet50 accuracy; (**b**) VGG16 accuracy; (**c**) MobileNet accuracy.

Table 3 shows a comparison of the evaluation indicators of ResNet50, VGG16 and MobileNet before and after using the transfer learning strategy in turn. It is obvious from the table that after the introduction of transfer learning, the evaluation indicators (Precision, Recall, F1-score and Accuracy) of each classification model have improved significantly, especially the Accuracy of VGG16 trained from scratch which was only 42.60% and the Precision which was only 44.65%. However, after the introduction of transfer learning pre-training the Accuracy reached 84.39% and the Precision reached 84.70%, displaying improvements of 41.79% and 40.05%, respectively.

**Table 3.** Comparison of model evaluation indicators before and after using Transfer learning.

| Model | Transfer Learning | Pre (%) | Rec (%) | F1 (%) | Acc (%) |
|---|---|---|---|---|---|
| ResNet50 | No | 73.74 | 72.10 | 72.70 | 73.99 |
| | Yes | 83.41 | 82.95 | 83.02 | 83.17 |
| VGG16 | No | 44.65 | 38.23 | 39.47 | 42.60 |
| | Yes | 84.70 | 83.69 | 84.09 | 84.39 |
| MobileNet | No | 67.85 | 66.22 | 66.61 | 69.58 |
| | Yes | 83.19 | 82.43 | 82.62 | 82.89 |

A complete comparison of the experimental results before and after the introduction of transfer learning for ResNet50, VGG16 and MobileNet is shown in Figure 9. As can be seen from the figure, based on using the same dataset IP_RicePests, the three classification networks of ResNet50, VGG16 and MobileNet were trained from scratch to obtain 73.99%, 42.60% and 69.58% accuracy, respectively, while the introduction of transfer learning with fine-tuned parameters was able to obtain 83.17%, 84.39%, and 82.89% accuracy, respectively, with accuracy improvements of 9.18%, 41.79% and 13.31%, respectively. It is demonstrated that the performance of all three models improved after the introduction of transfer learning. It is worth noting that the VGG16 classification model performs best in all evaluation indicators compared to the other two models, and its accuracy reached 84.39%.

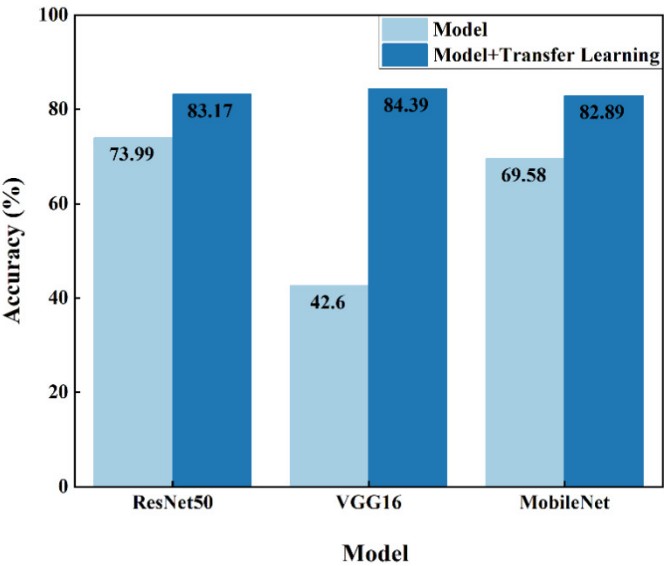

**Figure 9.** Accuracy comparison between each model before and after the introduction of transfer learning.

On the ResNet50 and MobileNet network classification models, although the network models can obtain lower initial training loss and validation loss during the training process after the introduction of transfer learning, the overfitting phenomenon is slightly more severe after transfer learning than the network models trained from scratch. However, the point is that the validation accuracy of the network model with transfer learning is always higher than that of the network model trained from scratch. This experiment also demonstrates that when the network size is large, the introduction of a transfer learning mechanism can effectively alleviate the overfitting phenomenon and can improve the recognition accuracy of the network for small pest samples.

### 4.2. Evaluation of the IP_RicePests Dataset

The IP102 paper states that this dataset demonstrates the best classification performance on the ResNet50 model, thus in this experiment, we also used the ResNet50 model to evaluate the effectiveness of the IP_RicePests dataset. The classification results under the same condition of using transfer learning are shown in Table 4.

**Table 4.** Performance comparison between IP102 and IP_RicePests on ResNet50.

| Datasets | Pre (%) | Rec (%) | F1 (%) | Acc (%) |
|---|---|---|---|---|
| IP102 | 65.83 | 64.85 | 65.10 | 65.35 |
| IP_RicePests | 83.41 | 82.95 | 83.02 | 83.17 |

From Table 4, we can see that IP_RicePests shows significant improvement in various evaluation indicators (Precision, Recall, F1-score and Accuracy) compared to IP102. For example, the accuracy of IP_RicePests on the ResNet50 classification model is 83.17%, compared to the accuracy of 65.35% for the rice dataset in IP102 on ResNet50, an overall improvement of 17.82%.

This experiment analyzed the classification accuracy of the IP102 dataset and IP_RicePests dataset on the ResNet50 model for each class of pests, as shown in Figure 10. Overall, it seems that for the classification accuracy of each class of pests, the Rice Pests dataset demonstrates a substantial improvement compared to the IP02 dataset. For example, the classification accuracy of the Rice leaf hopper in the IP102 dataset is only 61.48%, but its classification accuracy in IP_RicePests reaches 92.68%, revealing a direct improvement of 31.2%. The experimental results demonstrate that our proposed IP_RicePests dataset is a more effective dataset for rice pest classification.

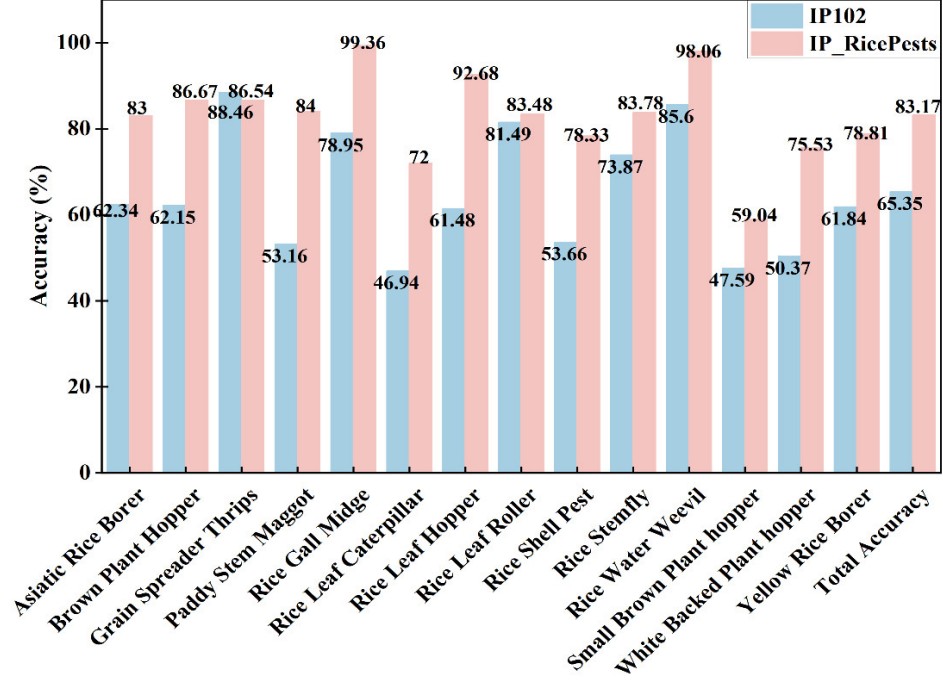

**Figure 10.** Classification accuracy of the ResNet50 model under the same condition of using transfer learning for 14 rice significant pests on the IP102 and IP_RicePests datasets, respectively. Total Accuracy signifies the total classification accuracy for the 14 rice significant pest categories.

### 4.3. ARGAN Data Augmentation Evaluation

To solve the problems of inter-class imbalance and large intra-class variation in the IP_RicePests dataset, we used ARGAN to perform data augmentation on the IP_RicePests

dataset to try achieve a balanced number of samples. The sample numbers of 14 classes of significant rice pests before and after enhancement are shown in Figure 11 below. Specifically, we used ARGAN data augmentation to expand the sample size for classes with a small sample size based on the IP_RicePests dataset, so that the sample size of all 14 classes reached 1000. This was initially performed to solve the problem of sample imbalance between classes in the IP_RicePests dataset. As can be seen from Figure 11, the grain spreader contains only 174 sample images before ARGAN data augmentation, and the number of samples reaches 1000 after the data augmentation.

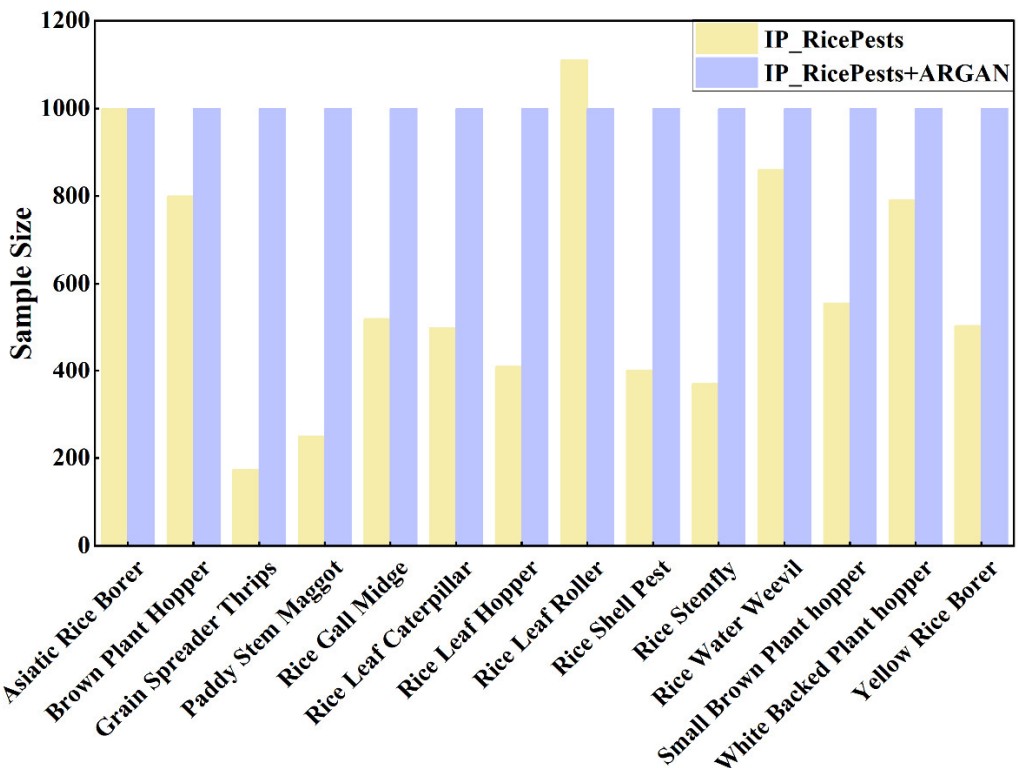

**Figure 11.** Comparison of sample numbers before and after ARGAN data augmentation.

The effect on some samples after data augmentation is shown in Figure 12. The top-left image is the original image, and the remaining 14 images are the new sample images generated after the original image has learned 14 types of pest characteristics. From the figure, the effectiveness of ARGAN data augmentation is obvious.

The enhanced dataset (samples from IP_RicePests and samples generated by ARGAN) was also divided into a training set, validation set and test set at a ratio of 6:2:2, and the ResNet50, VGG16 and MobileNet network models were trained separately based on transfer learning. Table 5 shows a comparison of the evaluation indicators of the three models before and after enhancement with ARGAN data augmentation. As can be seen from the table, the IP_RicePests dataset has some improvement in various evaluation metrics (Precision, Recall, F1-score and Accuracy) after solving the inter-class imbalance problem via ARGAN data augmentation. Specifically, after using ARGAN data augmentation on the IP_RicePests dataset, ResNet50, VGG16 and MobileNet obtained 87.41%, 88.68% and 86.44% accuracy, respectively, which were improved by 4.24%, 4.29% and 3.55%, respectively, compared with the original dataset. This proves that the accuracy of all three models improved after the number of samples of each type is balanced via ARGAN data augmentation.

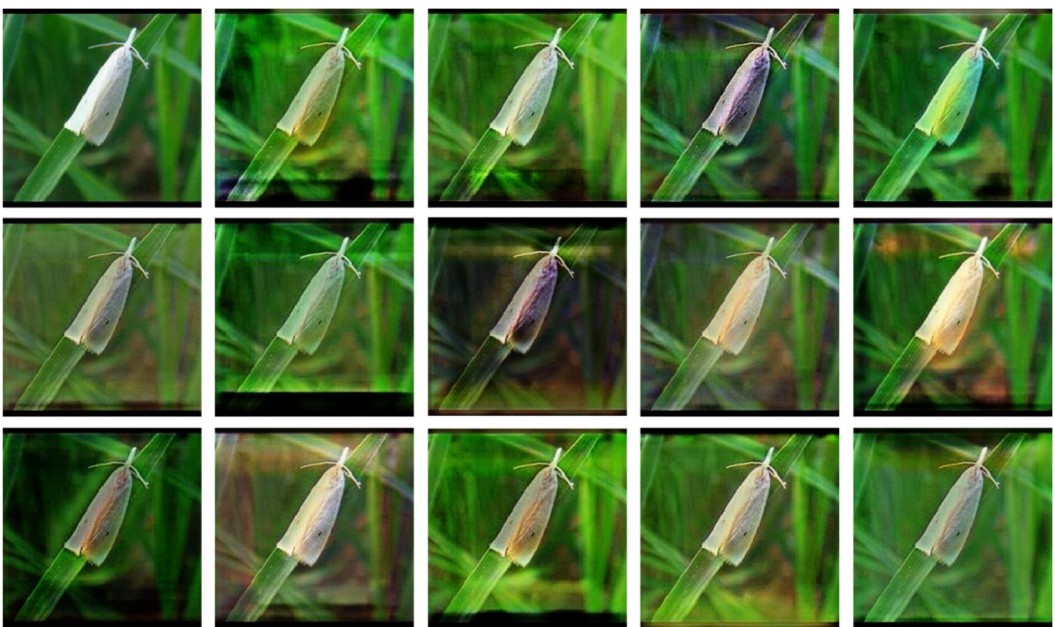

**Figure 12.** Results after using ARGAN data augmentation for Asiatic Rice Borer.

**Table 5.** Comparison of model evaluation indicators before and after using the ARGAN method.

| Model | ARGAN | Pre (%) | Rec (%) | F1 (%) | Acc (%) |
|---|---|---|---|---|---|
| ResNet50 | No | 83.41 | 82.95 | 83.02 | 83.17 |
| | Yes | 87.81 | 87.40 | 87.35 | 87.41 |
| VGG16 | No | 84.70 | 83.69 | 84.09 | 84.39 |
| | Yes | 88.87 | 88.68 | 88.69 | 88.68 |
| MobileNet | No | 83.19 | 82.43 | 82.62 | 82.89 |
| | Yes | 86.73 | 86.41 | 86.22 | 86.44 |

Figures 13–15 below show the classification accuracy of the ResNet, VGGNet and MobileNet classification models for each pests type before and after using the ARGAN data augmentation. It is obvious from the figures that the total classification accuracy of each model has basically improved. Firstly, as shown in Figure 13, the total classification accuracy of the IP_RicePests dataset after applying ARGAN data augmentation on the ResNet50 classification model was 87.41%, which is an improvement of 4.24% compared to that before enhancement. Specifically, the number of grain spreader thrips with the lowest original sample size was 174 before ARGAN data augmentation, and the classification accuracy was 86.54%, while the number of samples reached 1000 after data augmentation, and the accuracy also reached 98.33%, which demonstrates a significant increase of 11.79% compared with that before enhancement. Additionally, the number of original samples of paddy stem maggot, which was relatively small, also increased from 250 to 1000 after ARGAN data augmentation, and the classification accuracy increased from 84% to 91.33%, a direct improvement of 7.33% compared with that before ARGAN. Secondly, also on the VGG16 classification model (See Figure 14), the grain spreader thrips with minimal original samples exhibited a significant improvement in accuracy from 86.54% to 98% after ARGAN data augmentation. Finally, on the MobileNet classification model (see Figure 15), the accuracy of grain spreader thrips with the least original samples significantly improved from 92.31% to 99.67% after ARGAN data enhancement. Overall, the classification accuracy of the model is greatly improved after implementing ARGAN data augmentation, proving the effectiveness of ARGAN data augmentation for solving the problem of inter-class imbalance in the dataset.

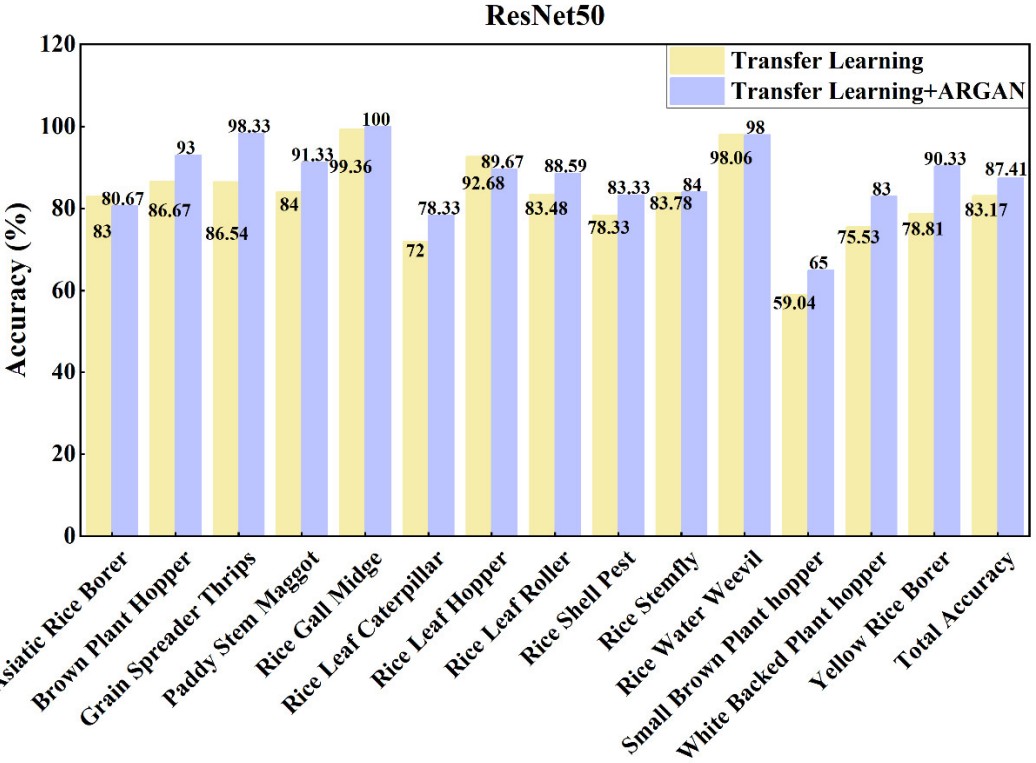

**Figure 13.** Comparison of classification accuracy of various pests by ResNet50 classification model before and after ARGAN data augmentation.

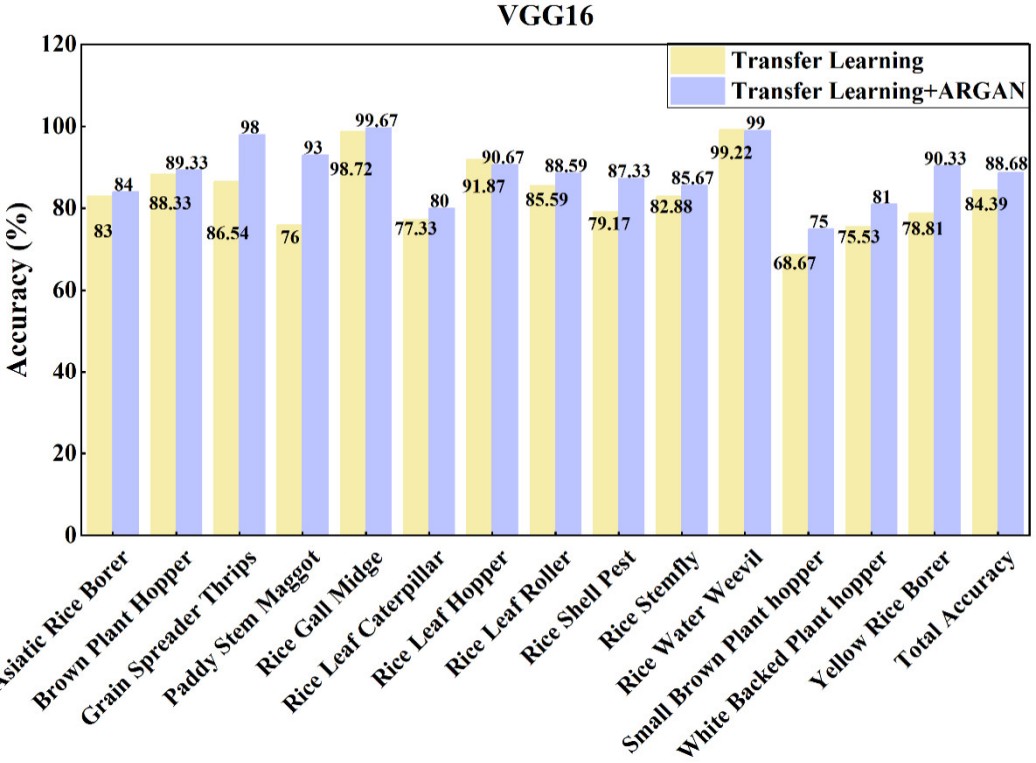

**Figure 14.** Comparison of classification accuracy of various pests by VGG16 classification model before and after ARGAN data augmentation.

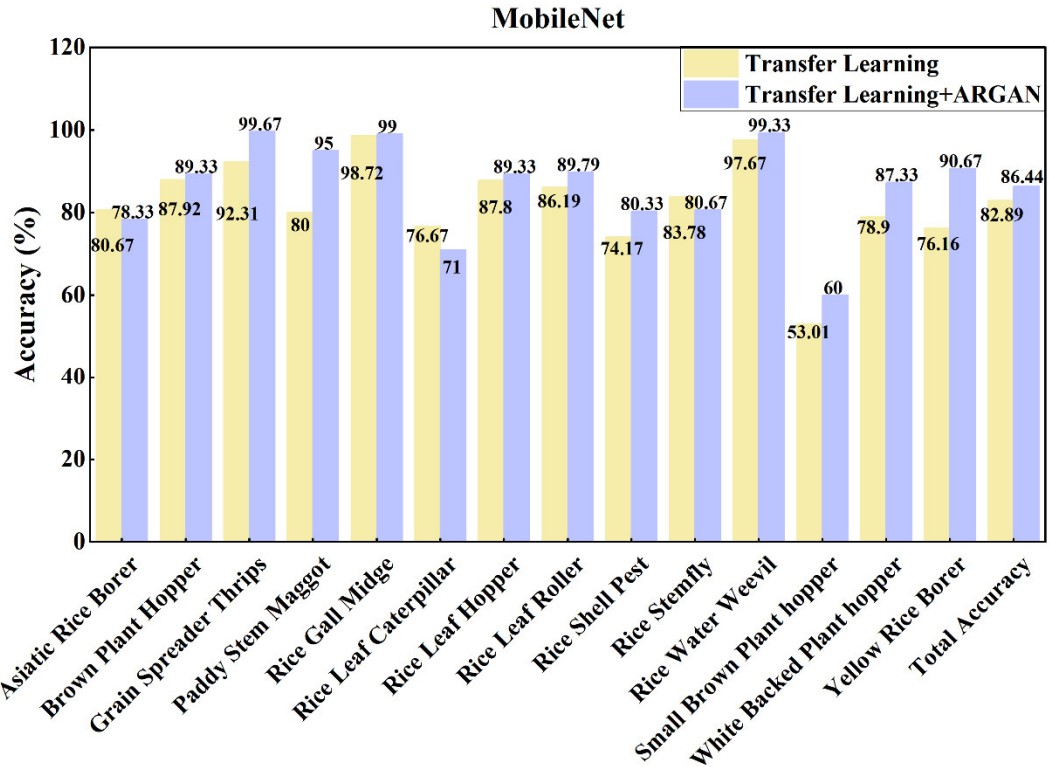

**Figure 15.** Comparison of classification accuracy of various pests by MobileNet classification model before and after ARGAN data augmentation.

## 5. Discussion

To mitigate the yield reduction caused by rice pest attacks on the stems, leaves and roots of rice plants, we used deep learning image classification models to accurately identify rice pests. Moreover, with regard to deep learning network training, the more the sufficient samples of the model are trained, the more generalized and robust the network model becomes.

According to the study, rice pest data resources are seriously insufficient and the accuracy of classification models in existing open-source datasets is generally not high. For example, the largest existing open-source pest classification dataset, IP102, has more room for image quality improvement (as shown in Figure 1). Therefore, we have performed the following:

1. Firstly, we hand-selected a total of 8417 images from 14 categories of rice pest samples in IP102 for two weeks, creating a higher quality rice pest data sample 1.0;
2. Secondly, we used crawler technology to expand a large number of images for each category of pest samples, and obtained a higher quality rice pest data sample 2.0 after hand-screening;
3. Thirdly, after reviewing the literature and receiving expert guidance, we manually screened the rice pest data sample 2.0 again, and finally formed a large-scale dataset on rice major pest classification, IP_RicePests.

While the sample numbers of IP_RicePests and IP102 remained basically the same (see Table 1), the accuracy of the same classification network using ResNet50 demonstrated a large difference (see Section 4.2): The accuracy of the IP_RicePests dataset at ResNet50 is 83.17%, displaying a 17.82% improvement compared to the accuracy of the IP102 dataset at ResNet50 measuring 65.35%. These findings confirm IP_RicePests currently as one of the most effective and open source datasets for the task of rice pest classification.

In the process of designing the experiments, we found that transfer learning is an effective way to optimize the model training process. Cao et al. [45], when conducting

the identification of common insects in the field, introduced a pre-trained model of the ImageNet dataset and then transferred the results to their model and obtained 97.39% recognition accuracy. This provided us a source of great inspiration. Therefore, in this experiment, we also chose the most common and widely used ImageNet dataset to pre-train the classification model, and transferred the learning results to the model using the IP_RicePests dataset. The results showed that the network after introduction of a transfer learning mechanism can effectively alleviate the overfitting phenomenon and improve classification accuracy of the model for small samples of pests.

IP_RicePests dataset suffers from the problem of inter-class number imbalance (see Table 1 and Figure 3) due to difficulty in collecting pictures of individual rice major pest samples in the context of real rice fields, such as grain spreader thrips, paddy stem maggot and rice stemfly. In addition, related studies have shown ARGAN to be an effective method in solving the problem of imbalance between classes in the dataset [27]. Therefore, in this experiment, we decided to use the ARGAN data augmentation method and expand the enhanced results to pest categories with fewer sample numbers in IP_RicePests to correct the problem of imbalance. The number of samples of 14 significant rice pest types before and after using ARGAN data augmentation is shown in Figure 11. The experimental results show that the Precision, Recall, F1-score and Accuracy of the model after using ARGAN data augmentation on the IP_RicePests dataset were greatly improved, and the inter-class imbalance problem of IP_RicePests was solved to an extent.

In general, in the context of the generally low accuracy of classification models based on existing open-source rice pest datasets, the IP_RicePests dataset after balancing the number of samples of various major rice pests via the ARGAN data augmentation method demonstrates excellent performance in the three common classification models based on transfer learning. In doing so, the need for accurate rice pest classification was successfully achieved in rice field scenarios. It is worth noting that the VGG16 classification model demonstrates the best performance in all evaluation metrics, with the total accuracy reaching 88.68%.

## 6. Conclusions

In this study, we proposed a large-scale dataset, namely IP_RicePests, applied to rice pest classification in response to the scarcity of resources regarding existing datasets. Firstly, we adopt a transfer learning approach by transferring model weights already trained on ImageNet to three classification network models: ResNet50, VGG16 and MobileNet, which are deep models helpful in rice pest identification. Compared with the other two models, the VGG16 classification model combined with transfer learning performed the best in all evaluation metrics, with an accuracy of 84.39%. To address the problem of inter-class imbalance and large intra-class variation in the IP_RicePests dataset, we performed ARGAN data augmentation on the IP_RicePests dataset. Compared with the other two models, the VGG16 classification model with Transfer Learning and ARGAN method performed the best in all evaluation metrics, and the accuracy of the model reached 88.68% on the IP_RicePests augmented dataset (samples from IP_RicePests and samples generated by ARGAN). The experiments show that CNN combined with transfer learning can employ the ARGAN data augmentation technique to overcome the difficulties in obtaining large samples and to improve the efficiency of rice pest identification. It is hoped that this study can provide foundational data and technical support for rice pest identification.

With respect to real application scenarios, this study still bears much room for improvement since the real-time captured images and the high-quality training samples contain some differences in features such as size, shape and color of rice pests.

For our future work, we will continue to collect high-quality pictures of major rice pests from real application scenarios using insect detection lights, and gradually improve the generalization ability of the classification model in real application scenarios. In addition, we will try to modify the structure of the network model VGG16 to further improve the classification accuracy and speed of the model.

**Author Contributions:** Conceptualization, X.J. (Xueqin Jiang) and Z.L.; methodology, X.J. (Xinyu Jia); software, X.J. (Xueqin Jiang) and X.J. (Xinyu Jia); validation, X.D., X.J. (Xinyu Jia) and Z.L.; formal analysis, X.J. (Xueqin Jiang) and Z.L.; investigation, X.J. (Xueqin Jiang) and X.D.; resources, J.M. and Z.L.; data curation, X.J. (Xinyu Jia) and X.J. (Xueqin Jiang); writing—original draft preparation, X.J. (Xueqin Jiang) and X.J. (Xinyu Jia); writing—review and editing, X.J. (Xueqin Jiang), J.M. and Z.L.; visualization, X.J. (Xinyu Jia) and Z.L.; supervision, J.M., Y.W. and Z.L.; project administration, Y.W. and Z.L.; funding acquisition, Y.W. and Z.L. All authors have read and agreed to the published version of the manuscript.

**Funding:** This research was funded by Research on intelligent monitoring and early warning technology for major rice pests and diseases of Sichuan Provincial Department of Science and Technology, grant number 2022NSFSC0172; Research and application of key technologies for intelligent spraying based on machine vision (key technology research project) of Sichuan Provincial Department of Science and Technology, grant number 22ZDYF0095.

**Data Availability Statement:** The data in this study are available on request from the corresponding author.

**Acknowledgments:** We would like to thank Luyu Shuai and Peng Cheng for their help in programming.

**Conflicts of Interest:** The authors declare no conflict of interest.

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
