# Peer review of "Classification Method of Significant Rice Pests Based on Deep Learning"

_agronomy, doi:10.3390/agronomy12092096_

Round 1

Reviewer 1 Report

Summary

In this paper, the authors proposed a new large scale dataset that dedicated for rice pests, and trained deep learning models on top of it to achieve better prediction performance. The author also claimed that transfer learning and data augmentation techniques can further boost the model performance.

Comments

1. In section 3.1, line 124 should not be there.

2. In figure 5-7, why is there a sudden drop in the loss curve at around 100 epoch? And the same question for figure 8.

3. Are the generated samples by ARGAN included in the IP_RicePests dataset?

Reviewer 2 Report

This study reports a large-scale dataset IP_RicePests applied to rice pest 465 classification against the background of scarce resources of existing datasets. Finally the VGG16 classification model combined with transfer learning was found performed the best with the accuracy of the model reached 84.39%.  Also, the VGG16 classification model with Transfer 474 Learning and ARGAN method performed the best on all evaluation metrics, and the 475 accuracy of the model reached 88.68%. That would be facilitating the rice pest control in the near future. Researchers relating to this area would be pretty interested in this article.

However, the organization and Englished language presentation are poor. For example, Figure 1 incorporated non-English figure descriptions. English grammar errors can be seen all over the manuscript. For instance, on line 276, it presented "Figure 8 display the accuracy curves of the validation sets...". It should be "Figure 8 displays/displayed". On line 428, "while the sample numbers of IP_RicePests and IP102 remained basically the...", While should be with capital W. Therefore, English language errors must be corrected.

The manuscript discussed that Precision, Recall, F1-score and Accuracy of the model were improved, but haven't discussed the still-existing shortcomings and how to improve in the future.
